

# Genetic structure and diversity of the selfing model grass *Brachypodium stacei* (Poaceae) in Western Mediterranean: out of the Iberian Peninsula and into the islands

Valeriia Shiposha[1,2], Pilar Catalán[1,2], Marina Olonova[2] and Isabel Marques[1]

[1] Department of Agriculture and Environmental Sciences, High Polytechnic School of Huesca, University of Zaragoza, Huesca, Spain
[2] Department of Botany, Institute of Biology, Tomsk State University, Tomsk, Russia

Corresponding author
Isabel Marques, isabel.ic@gmail.com

## ABSTRACT

Annual Mediterranean species of the genus *Brachypodium* are promising model plants for energy crops since their selfing nature and short-life cycles are an advantage in breeding programs. The false brome, *B. distachyon*, has already been sequenced and new genomic initiatives have triggered the de-novo genome sequencing of its close relatives such as *B. stacei*, a species that was until recently mistaken for *B. distachyon*. However, the success of these initiatives hinges on detailed knowledge about the distribution of genetic variation within and among populations for the effective use of germplasm in a breeding program. Understanding population genetic diversity and genetic structure is also an important prerequisite for designing effective experimental populations for genomic wide studies. However, population genetic data are still limited in *B. stacei*. We therefore selected and amplified 10 nuclear microsatellite markers to depict patterns of population structure and genetic variation among 181 individuals from 19 populations of *B. stacei* occurring in its predominant range, the western Mediterranean area: mainland Iberian Peninsula, continental Balearic Islands and oceanic Canary Islands. Our genetic results support the occurrence of a predominant selfing system with extremely high levels of homozygosity across the analyzed populations. Despite the low level of genetic variation found, two different genetic clusters were retrieved, one clustering all SE Iberian mainland populations and the island of Minorca and another one grouping all S Iberian mainland populations, the Canary Islands and all Majorcan populations except one that clustered with the former group. These results, together with a high sharing of alleles (89%) suggest different colonization routes from the mainland Iberian Peninsula into the islands. A recent colonization scenario could explain the relatively low levels of genetic diversity and low number of alleles found in the Canary Islands populations while older colonization events are hypothesized to explain the high genetic diversity values found in the Majorcan populations. Our study provides widely applicable information about geographical patterns of genetic variation in *B. stacei*. Among others,

the genetic pattern and the existence of local alleles will need to be adequately reflected in the germplasm collection of *B. stacei* for efficient genome wide association studies.

# INTRODUCTION

Approximately one third of Earth's land is covered by grass-dominated ecosystems comprising 600 genera and more than 12,000 species (*Soreng et al., 2015*). Besides their important ecological role, grasses are the core of human nutrition and several genomic efforts have focused on economically important species (e.g., rice: *International Rice Genome Sequencing Project (2005)*; sorghum: *Paterson et al. (2009)*). Among grasses, the genus *Brachypodium*, a member of the Pooideae subfamily, has recently been developed as a new model system to study the evolution of grasses. The genome of the annual *B. distachyon*, commonly known as the false brome, has already been sequenced (*International Brachypodium Initiative, 2010*). This species has several features suitable for the development of a model plant for genomic studies such as a small diploid genome (~355 Mbp), a short annual life-cycle, easily amenable to culture, and a selfing nature (*Gordon et al., 2014*).

The taxonomic identity of *B. distachyon* was recently challenged with the recognition that the three cytotypes attributed to different ploidy levels in this species (e.g., an autopolyploid series of individuals with x = 5 and 2n = 10 (2x), 20 (4x), 30 (6x) chromosomes; *Robertson, 1981*) were in fact three different species: two diploids, each with a different chromosome base number, *B. distachyon* (x = 5, 2n = 10) and *B. stacei* (x = 10, 2n = 20), and their derived allotetraploid *B. hybridum* (x = 5 + 10, 2n = 30) (*Catalán et al., 2012*; *López-Alvarez et al., 2012*). This recent taxonomic split has triggered new genomic initiatives including the re-sequencing of 56 new accessions of *B. distachyon* and the de-novo genome sequencing of *B. stacei* and *B. hybridum*, a project undertaken by the Joint Genome Institute and the International *Brachypodium* Consortium (http://jgi.doe.gov/our-science/science-programs/plant-genomics/brachypodium/). The forthcoming genomes of *B. stacei* and *B. hybridum* will allow the development of several functional genomic analyses on these diploid and polyploid species and their potential transfer to other cereals and forage or biofuel crops. A recent update on phenotypic traits and habitat preferences of the three species has increased the number of discriminant features that distinguish them and has thrown new insights into their respective ecological adaptations (*Catalán et al., 2016a*). However, very scarce genetic information exists for these close relatives of *B. distachyon*, especially for the rarest species of this complex, *B. stacei* (*Catalán et al., 2016b*). It would, therefore, be invaluable to have more information especially because a collection of germplasm reflecting the natural diversity of *B. stacei* is necessary for future genome wide association studies and the creation of reference lines.

*Brachypodium stacei* is a monophyletic annual diploid species that diverged first from the common *Brachypodium* ancestor, followed consecutively by *B. mexicanum*, *B. distachyon* and the clade of the core perennial taxa (*Catalán et al., 2012*; *Catalán et al., 2016b*). Several studies have revealed it to be distinct from *B. distachyon* and *B. hybridum*: e.g., protein data: *Hammami et al. (2011)*; nuclear SSRs: *Giraldo et al. (2012)*; DNA barcoding: *López-Alvarez et al. (2012)*; isozymes: *Jaaska (2014)*. A recent study using environmental niche models predicted a potential distribution of *B. stacei* in coastal and lowland areas of the circum-Mediterranean region (*López-Alvarez et al., 2015*), concurrent with its known geographic distribution (*Catalán et al., 2016a*). However, a large number of those populations occur in the western Mediterranean region and in Macaronesia (*López-Alvarez et al., 2015*; *Catalán et al., 2016a*). Population genetic studies conducted in its annual congener *B. distachyon* have demonstrated that the genetic structure does not fit a geographic pattern but rather might have resulted from a combination of factors such as long distance dispersal of seeds and flowering time isolation (*Vogel et al., 2009*; *Mur et al., 2011*; *Tyler et al., 2016*).

Here, we studied the patterns of genetic variation in the mainland Iberian Peninsula and the western island populations (continental Balearic Islands and oceanic Canary Islands) of *B. stacei* to unravel the origin and phylogeographic patterns of its populations. From all its range, this area is the best known due to previous studies (*Catalán et al., 2012*; *Catalán et al., 2016a*; *López-Alvarez et al., 2012*; *López-Alvarez et al., 2015*), which can guarantee the correct identification of *B. stacei* since it can be misidentified with its close-relatives (*López-Alvarez et al., 2012*). We specifically addressed the following questions: (1) Is genotypic diversity within populations limited by the prevalence of autogamous pollinations? (2) Do islands (e.g., continental, oceanic) contain less genetic variation than mainland areas? (3) Is there a signature of geographic genetic structure in this self-pollinated plant? Finally, we aim to provide recommendations necessary to establish an efficient germplasm collection of *B. stacei*, with the aim of helping future genomic initiatives in *Brachypodium*.

## MATERIAL AND METHODS

### Population sampling, DNA extraction and nSSR amplification

A total of 181 individuals were sampled from 19 populations of *B. stacei* covering the whole distribution range of this species within the Iberia Peninsula, plus the continental Balearic (Gymnesic) Islands (Majorca, Minorca) and the oceanic Canary Islands (Gomera, Lanzarote) (Table 1; Fig. 1). Nine populations were sampled in mainland Iberian Peninsula and ten across the two groups of islands (Fig. 1). In each population, ten individuals were collected randomly with a minimum sampling distance of 10 m, with the exception of the Iberian ALI and the Majorcan BANYA populations where only five and six individuals were respectively found. Sampling sizes, locations and geographic coordinates of each population sampled are given in Table 1. Fresh leaves were collected for each individual, dried in silica gel and stored at −20 °C until ready for DNA isolation. The silica samples for all individuals were deposited in the DNA bank of the BioFlora

**Table 1 Sampled populations of *Brachypodium stacei* sorted by geographical area.** The location, population code, number of plants genotyped (*N*), mean observed heterozygosity (*H$_o$*) and expected heterozygosity (*H$_e$*), mean number of alleles (*N$_a$*), allelic richness (*A$_R$*), inbreeding coefficient (*F$_{IS}$*), selfing rate (*s*), and number of exclusive genotypes (%. between parenthesis) are shown.

| Locality | Code | N | Latitude (N) | Longitude (W) | H$_o$ | H$_e$ | N$_a$ | A$_R$ | F$_{IS}$ | s | Exclusive genotypes |
|---|---|---|---|---|---|---|---|---|---|---|---|
| **Mainland (Iberian Peninsula)** | | | | | | | | | | | |
| S Spain: Granada, Moclin | GRA | 10 | 37°19′59″N | 3°46′59″W | 0.240 | 0.155 | 12 | 1.126 | 0.667* | 0.800 | 3 (30%) |
| S Spain: Almeria, Cabo de Gata | ALM | 10 | 36°44′2″N | 2°8′35″W | 0.170 | 0.102 | 11 | 1.050 | 0.0001 | 0.0001 | 3 (30%) |
| S Spain: Jaen: Cazorla, Cortijos Nuevos | JAE1 | 10 | 38°11′31″N | 2°48′14″W | 0.120 | 0.116 | 12 | 1.176 | 0.723* | 0.839 | 4 (40%) |
| S Spain: Jaen: Quesada, Tiscar | JAE2 | 10 | 37°46′5″N | 3°1′23″W | 0.200 | 0.100 | 10 | 1.000 | – | – | 1 (10%) |
| SE Spain: Murcia, Portman | PORT | 10 | 37°34′57″N | 0°51′15″W | 0.200 | 0.100 | 10 | 1.000 | – | – | 1 (10%) |
| SE Spain: Murcia, Calblanque | CALBN | 10 | 37°35′59″N | 0°45′29″W | 0.140 | 0.108 | 14 | 1.246 | 0.526* | 0.689 | 4 (40%) |
| SE Spain: Murcia, Cobaticas | CALBA | 10 | 37°35′59″N | 0°45′30″W | 0.110 | 0.105 | 15 | 1.339 | 0.617* | 0.763 | 5 (50%) |
| SE Spain: Murcia, Cala Reona | CALREL | 10 | 37°36′56″N | 0°42′56″W | 0.030 | 0.009 | 13 | 1.239 | 0.520* | 0.684 | 5 (50%) |
| SE Spain: Alicante, Cabo La Nao | ALI | 5 | 38°45′22″N | 0°13′8″E | 0.300 | 0.150 | 10 | 1.000 | – | – | 1 (20%) |
| **Balearic (Gymnesic) Islands** | | | | | | | | | | | |
| Spain: Minorca: Es Mercadal, Toro | MEN | 10 | 39°59′6″N | 4°6′47″E | 0.240 | 0.173 | 13 | 1.203 | 0.386* | 0.556 | 3 (30%) |
| Spain: Majorca: Sa Dragonera, Gambes | DRAG | 10 | 39°35′13″N | 2°19′37″E | 0.111 | 0.154 | 16 | 1.428 | 0.916* | 0.956 | 5 (50%) |
| Spain: Majorca: Arta, Peninsula de Llevant | ARTA | 10 | 39°44′10″N | 3°20′6″E | 0.210 | 0.128 | 12 | 1.126 | 0.666* | 0.799 | 3 (30%) |
| Spain: Majorca: Campanet, Coves | CAMPA | 10 | 39°47′31″N | 2°58′12″E | 0.130 | 0.138 | 14 | 1.434 | 0.486* | 0.654 | 6 (60%) |
| Spain: Majorca: Alcudia, Punta Negra | ALCU | 10 | 39°52′48″N | 3°10′41″E | 0.140 | 0.108 | 14 | 1.200 | 0.0001 | 0.0001 | 2 (20%) |
| Spain: Majorca: Felenitx, San Salvador | FELEN | 10 | 39°27′4″N | 3°11′17″E | 0.130 | 0.109 | 14 | 1.200 | 0.250 | 0.400 | 4 (40%) |
| Spain: Majorca: Petra, Bonany | BONA | 10 | 39°35′38″N | 3°5′10″E | 0.290 | 0.391 | 23 | 1.992 | 0.385* | 0.5555 | 9 (90%) |
| Spain: Majorca: Banyalbufar, Ses Animes | BANYA | 6 | 39°41′6″N | 2°30′36″E | 0.167 | 0.239 | 15 | 1.496 | 0.825* | 0.904 | 6 (100%) |
| **Canary Islands** | | | | | | | | | | | |
| Spain: Gomera: Agulo | GOM | 10 | 28°10′59″N | 17°10′59″W | 0.150 | 0.118 | 11 | 1.076 | 0.891* | 0.942 | 2 (20%) |
| Spain: Lanzarote: Teguise | LAN | 10 | 29°4′1″N | 13°31′1″W | 0.230 | 0.136 | 11 | 1.096 | 1.000* | 1 | 2 (20%) |

**Notes:**
* F$_{IS}$ values deviating from HWE (*P* > 0.05).

group at the University of Zaragoza in Spain and voucher specimens were deposited in the JACA herbarium (Spain).

Total genomic DNA was extracted from fresh leaf tissue or from silica-dried leaf samples using the DNeasy Plant Mini Kit (Qiagen, Valencia, CA, USA) according to the manufacturer's protocol. The 181 samples used in this study were genotyped at 10 variable nuclear microsatellite markers (nSSRs) developed for *B. distachyon* (*ALB006, ALB022, ALB040, ALB050, ALB086, ALB087, ALB139, ALB165, ALB181* and *ALB311*; Vogel et al., 2009). All those microsatellites were selected because during our preliminary studies they displayed good quality and high transferability success in *B. stacei*. The forward primer of each locus was 5-end labeled with a fluorescent dye. Amplifications were carried out in a final volume of 10 μl containing between 0.1 and 0.2 μl of each 10 m diluted primer (forward and reverse), 5 μl PCR Master Mix (QIAGEN) and 2.5 μl DNA. The polymerase chain reactions (PCR) were carried out on a GeneAmp PCR System 9700 thermocycler with a thermal profile consisting of a 4-min initial denaturation step at 95 °C followed by 35 cycles of 30 s at 95 °C, 30 s at 55 °C and 1 min at 72 °C. A final 72 °C extension step of
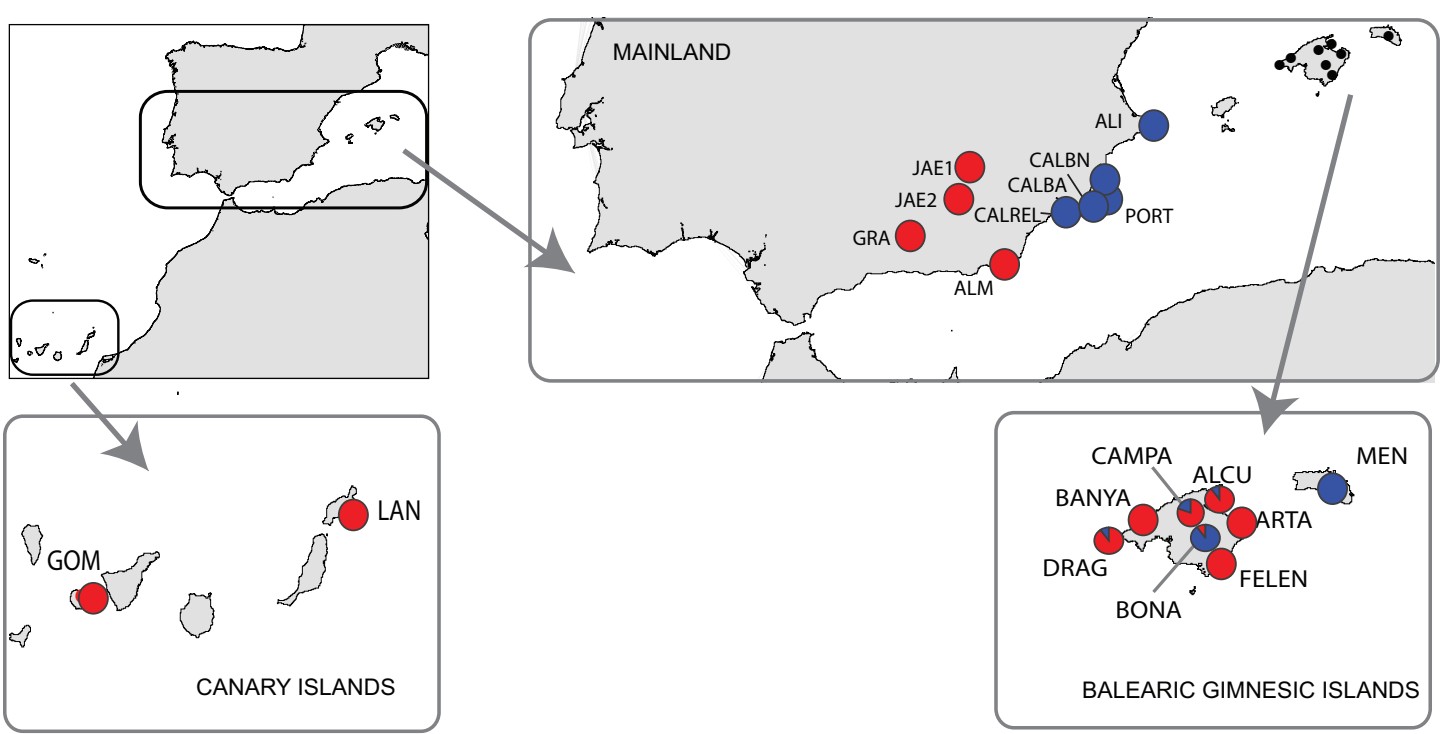

**Figure 1 Location of the study area of *Brachypodium stacei*.** Collection localities of *Brachypodium stacei* populations in mainland Iberian Peninsula, the continental Balearic (Gymnesic) Islands (Minorca and Majorca) and the oceanic Canary Islands. Pie-charts indicate the proportion of ancestry assigned to individuals of each population by Bayesian clustering analysis using STRUCTURE.

30 min was included to promote non-templated nucleotide addition at the 3′end of the PCR product. Multiplexed PCR products were genotyped on an Applied Biosystems 3130XL Genetic Analyzer using 2 μl of amplified DNA, 12 μl of Hi-Di formamide and 0.4 μl of GeneScan-500 (LIZ) size standard (Applied Biosystem). Allele sizes were determined using Peak Scanner version 1.0 (Life Technologies). Within each population, all loci were checked for the presence of null alleles using MICRO-CHECKER v.2.2.3 (*van Oosterhout et al., 2004*).

## Hardy-Weinberg equilibrium, linkage disequilibrium and genetic diversity

Deviation from Hardy-Weinberg Equilibrium (HWE) was tested at both population and locus levels using FSTAT 2.9.3.2 (*Goudet, 2001*). To calculate the extent of linkage disequilibrium between pairs of loci (LD) in each population we set dememorization numbers at 10,000 and performed 100,000 iterations for all permutation tests (exact tests) in Genepop v.4.0.10 (*Raymond & Rousset, 1995*). Significant values were corrected for multiple comparisons by Bonferroni correction (*Rice, 1989*).

For each microsatellite locus and population, genetic polymorphism was assessed by calculating the total number of alleles ($N_a$, allelic diversity), mean expected heterozygosity ($H_e$), mean observed heterozygosity ($H_o$), allelic richness ($A_R$), and inbreeding coefficient ($F_{IS}$) using FSTAT 2.9.3.2 (*Goudet, 2001*). The inbreeding coefficient was also estimated using the Bayesian procedure (IIM) implemented in INEst 2.0, which is

robust to the presence of null alleles (*Chybicki & Burczyk, 2009*). Posterior distribution was based on 300,000 steps, sampling every 100 steps and discarding the first 30,000 steps as burn-in. In order to infer the statistical significance of inbreeding we compared the full model (nfb), the model including only the possibility of null alleles and inbreeding (nf), and the model including only null alleles and genotyping failures (nb). The best model was chosen based on the Deviance Information Criterion (DIC; cf. *Chybicki, Oleksa & Burczyk, 2011*).

GenAlEx 6 software was used to estimate the mean expected heterozygosity ($H_e$) and mean observed heterozygosity ($H_o$) for each population (*Peakall & Smouse, 2006*). In addition, the selfing rate ($s$) was also estimated as $s = 2F_{IS}/(1 + F_{IS})$ (*Ritland, 1990*). Spatial patterns of allelic quantity were visualized by mapping variation for the locations across space with the interpolation kriging function in ARCINFO (ESRI, Redlands, CA, USA), using a spherical semivariogram model.

## Population genetic structure, genetic differentiation and isolation

The Bayesian program STRUCTURE v.2.3.4 (*Pritchard, Stephens & Donnelly, 2000*) was used to infer the population structure and to assign individual plants to subpopulations. Models with a putative numbers of populations ($K$) from 1–10, imposing ancestral admixture and correlated allele frequencies priors, were considered. Ten independent runs with 50,000 burn-in steps, followed by run lengths of 300,000 interactions for each $K$, were computed. The number of true clusters in the data was estimated using STRUCTURE HARVESTER (*Earl & vonHoldt, 2012*), which identifies the optimal $K$ based both on the posterior probability of the data for a given $K$ and the $\Delta K$ (*Evanno, Regnaut & Goudet, 2005*). To correctly assess the membership proportions ($q$ values) for clusters identified in STRUCTURE, the results of the replicates at the best fit $K$ were post-processed using CLUMPP 1.1.2 (*Jakobsson & Rosenberg, 2007*). BAPS v.5.2 (*Corander, Marttinen & Mäntyniemi, 2006*) was used to explore population structure further. In contrast to STRUCTURE, BAPS determines optimal partitions for each candidate $K$-value and merges the results according the log-likelihood values to determine the best $K$-value. Analyses in BAPS were done at the level group of individuals using the models without spatial information and by selecting 1–10 as possible $K$-values. Ten repetitions were performed for each $K$. POPULATION 1.2 (*Langella, 2000*) was used to calculate the Nei's genetic distance ($D_A$; *Nei, Tajima & Tateno, 1983*) among individuals and to construct an unrooted neighbor-joining tree with 1,000 bootstrap replicates. Nei's genetic distance among individuals was also visualized by Principal Components Analysis (PCoA) with GenAlEx6 (*Peakall & Smouse, 2006*).

We estimated genetic differentiation among locations using an analysis of molecular variance (AMOVA) with ARLEQUIN 3.11 (*Excoffier, Laval & Schneider, 2005*). In addition, molecular variance was also studied (1) between the genetic groups retrieved by STRUCTURE and BAPS, (2) between mainland and island populations, (3) within mainland populations, e.g., S Spain vs. SE Spain, and (4) within island populations, e.g., Balearic vs. Canary Islands. In each analysis, variance was quantified among groups, among locations within groups and within sampling locations. Each AMOVA was run

with 10,000 permutations at 0.95 significance levels. Relationships between genetic and linear geographic distances (isolation-by-distance, IBD) were examined using a Mantel test (*Mantel, 1967*) implemented in ARLEQUIN 3.11 (*Excoffier, Laval & Schneider, 2005*) with 10,000 permutations.

## RESULTS

### Hardy-Weinberg disequilibrium, non linkage disequilibrium

Deviations from HWE were common in the selfed *B. stacei*. From the 19 populations sampled, only five were at HWE (GRA, MEN, ARTA, FELEN, GOM) at the 5% level after the sequential Bonferroni correction (Table 2). Pairwise comparisons between loci revealed no significant linkage disequilibrium at the $P = 5\%$ suggesting that alleles are assorting independently at different loci.

### Genetic diversity and mating system of *Brachypodium stacei*

For each locus, observed heterozygosity values ranged from 0 to 0.058 (respectively for loci *ALB139* and *ALB086*), and expected heterozygosity ranged from 0 to 0.145 (respectively for loci *ALB139* and *ALB087*). $F_{IS}$ values varied between −0.068 and 0.8482 (respectively for loci *ALB311* and *ALB022*; Table 3) across the loci studied. No null alleles were detected. The results from the Bayesian analyses implemented in INEst revealed that only inbreeding contributed to the excessive homozogosity, since this model ($DIC_{nf}$: 3,300.019) was preferred over the alternative ones ($DIC_{nfb}$: 4,400.390; $DIC_{nb}$: 4,400.300) based on the DIC criterion.

From the 19 sampled populations of *B. stacei*, only 37 distinct alleles were found in the 181 individuals studied (Fig. 2; Table S1). Most of the alleles (27 alleles; 73%) were shared between populations while the remaining ones were private to mainland, Majorca, Minorca or the Canary Islands (10 alleles; 27%). Most of the alleles found in the islands were also found in the mainland since only three alleles out of 27 (11%) were not found in the mainland: two alleles were shared between Majorca and Minorca and one allele was shared between Majorca and the Canary Islands (Fig. 2). Out of 37, four alleles were exclusively found only in the mainland (10%; three in SE Spain and one in S Spain), six in Majorca (16%) and one in Minorca (2%) while the Canary Islands had no unique alleles (Fig. 2).

The number of alleles generally increased in the Balearic Islands, most specially in Majorca ($P < 0.0001$) as shown when projected into the geographic space (Fig. 3). Overall, only 38% (69 out of 181) of all genotyped samples exhibited unique multi-locus genotypes, as a consequence of the rampant homozygosis (fixed alleles) observed for most loci in most populations. The observed percentage is lower than one might expect under random mating, where the frequency of multilocus genotypes is expected to be equal to the product of the allelic frequencies. However, a relatively high number of unique multi-locus genotypes were generally found in the populations collected in the island of Majorca, where up to 100% of all the individuals sampled showed unique multi-locus genotypes (Table 1).

Mean observed heterozygosity among the populations of *B. stacei* varied between 0.110 (mainland population CALBA) and 0.290 (Majorcan population BONA) with a

**Table 2 Results of the Hardy Weinberg exact tests retrieved by GENEPOP for 19 populations of Brachypodium stacei.** *P*-value (0.05) associated with the null hypothesis of random union of gametes (or '−' if no data were available, or only one allele was present) estimated with a Markov chain algorithm and the standard error (S.E.) of this estimate.

| Population | P-value | S.E. |
| --- | --- | --- |
| GRA | 0.0519 | 0.0011 |
| ALM | − | |
| JAE1 | 0.0259 | 0.0007 |
| JAE2 | − | |
| PORT | − | |
| CALBN | 0.0077 | 0.0009 |
| CALBA | 0.0007 | 0.0001 |
| CALREL | 0.0249 | 0.0006 |
| ALI | − | |
| MEN | 0.1016 | 0.0015 |
| DRAG | 0 | 0 |
| ARTA | 0.053 | 0.0012 |
| CAMPA | 0.0361 | 0.0014 |
| ALCU | − | |
| FELEN | 0.0508 | 0.0028 |
| BONA | 0 | 0 |
| BANYA | 0 | 0 |
| GOM | 0.0515 | 0.0012 |
| LAN | 0.0096 | 0.0005 |

**Table 3 Characteristics and genetic diversity statistics of the nuclear microsatellite markers used in the genetic study of Brachypodium stacei.** For each locus, the total number of alleles ($N_a$), mean expected heterozygosity ($H_e$), mean observed heterozygosity ($H_o$), and the fixation index ($F_{IS}$) obtained from the 181 studied individuals are shown.

| Locus | Repeat motif | $N_a$ | $H_e$ | $H_o$ | $F_{IS}$ |
| --- | --- | --- | --- | --- | --- |
| ALB006 | (GT)15 | 2 | 0.016 | 0.016 | 0.003 |
| ALB022 | (CT)11 | 2 | 0.035 | 0.005 | 0.848 |
| ALB040 | (CTT)8 | 4 | 0.129 | 0.047 | 0.632 |
| ALB050 | (GT)15 | 4 | 0.122 | 0.032 | 0.717 |
| ALB086 | (AAG)7 | 6 | 0.119 | 0.058 | 0.486 |
| ALB087 | (AGC)7 | 6 | 0.145 | 0.032 | 0.758 |
| ALB139 | (AGA)7 | 1 | 0.000 | 0.000 | 0 |
| ALB165 | (ATA)12 | 4 | 0.066 | 0.049 | 0.298 |
| ALB181 | (AC)9 | 5 | 0.049 | 0.037 | 0.253 |
| ALB311 | (GA)6 | 3 | 0.025 | 0.026 | −0.069 |

CI of ± 0.03 at the 95% level, while mean expected heterozygosity varied between 0.090 (mainland population CALREL) and 0.239 (Majorcan population BANYA; Table 1) with a CI of ± 0.04 at the 95% level. The average $F_{IS}$ value was 0.558 (CI: 0.141) varying between 0.0001 (mainland population ALM, Majorcan population ALCU) and 1 (Canary

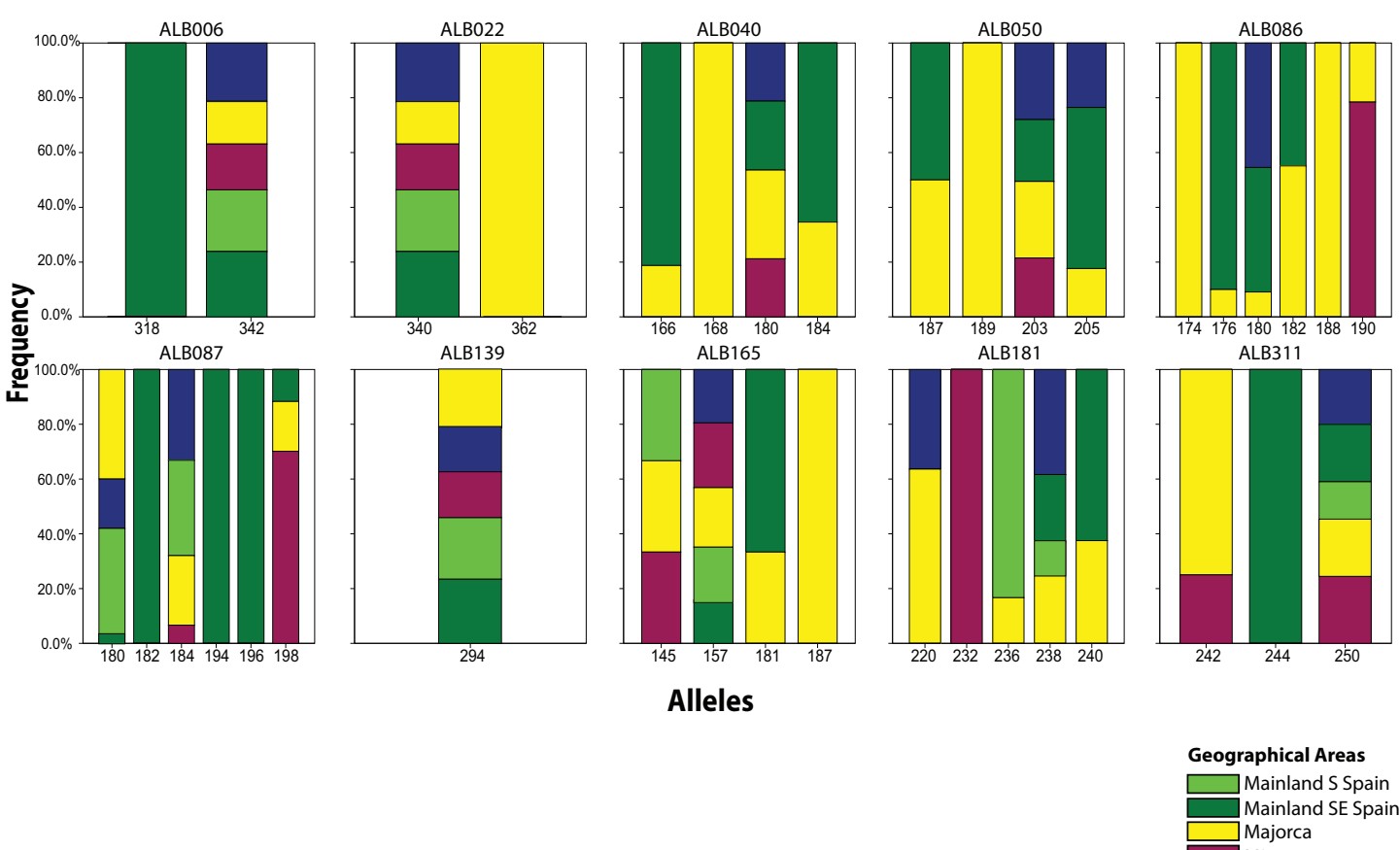

**Figure 2 Distribution of *Brachypodium stacei* alleles.** Frequency of the alleles found in *Brachypodium stacei* across the geographical area sampled: mainland Iberian Peninsula (SE Spain and S Spain) and the islands of Minorca, Majorca and the Canary Islands. Colors of areas are indicated in the chart.

population of LAN). Therefore, the average rate of self-fertilization in *B. stacei* was estimated to be 71% considering all the populations (Table 1). However, the wide range of $F_{IS}$ values implies that the predicted level of self-fertilization also varies extensively across populations with a CI of ± 0.25 at the 95% level.

## Population genetic structure among geographical areas

The Bayesian clustering program STRUCTURE found the highest LnP(D) and ΔK values for $K = 2$ ($P < 0.001$) which differentiated all south (S) Iberian mainland populations from the southeastern (SE) mainland populations. The populations collected in the island of Minorca clustered with the SE mainland populations, whereas samples from the Canary Islands and most of the Majorcan populations were grouped with the S mainland populations, with the exception of the Majorcan population of BONA were most individuals were assigned to the same genetic group found only in the SE mainland populations (Fig. 4A). Some individuals collected in four populations of Majorca showed genetic admixture between the two genetic groups (DRAG, CAMPA, ALCU, BONA; Fig. 4A). These results were also

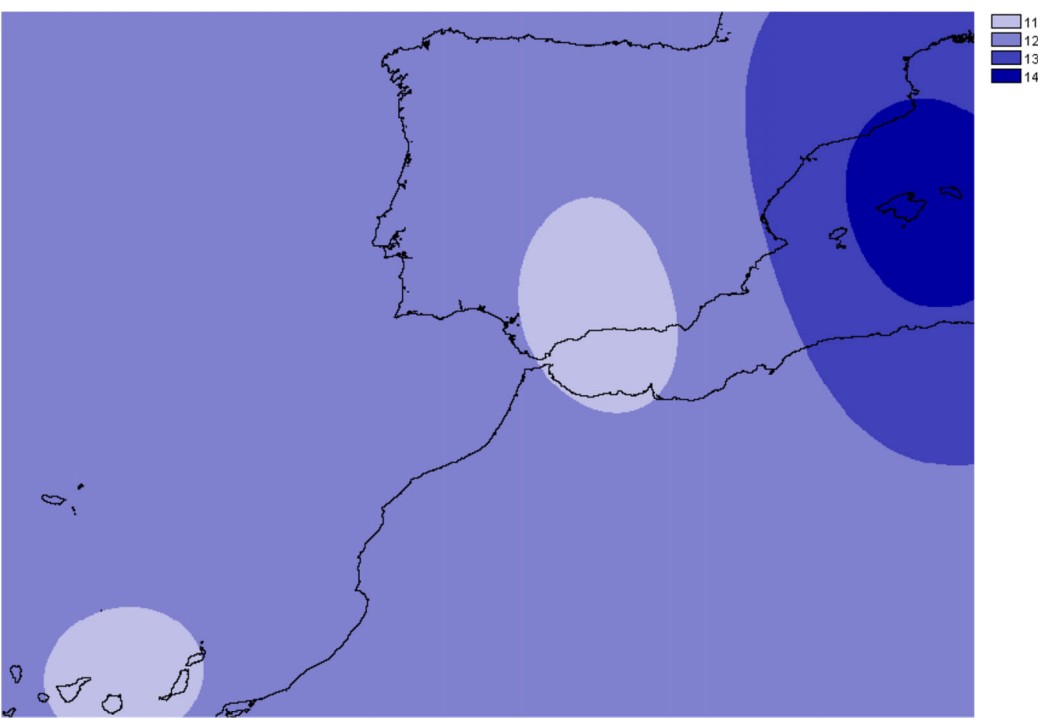

**Figure 3 Overall allelic richness of *Brachypodium stacei*.** Map of overall allelic richness of *Brachypodium stacei* across the geographic range sampled. Dark areas contain higher richness.

corroborated by BAPS, which retrieved similar results and generally differentiated S mainland populations, Canary Islands and Majorcan populations from all the remaining populations sampled with the exception of the Majorcan population of BONA (Fig. 4B).

The PCoA spatially separated SE Iberian mainland and the Minorca island populations from all remaining populations at both extremes of axis I, which accumulated 44.3% of variance (Fig. 5), partially supporting the genetic boundaries assigned by STRUCTURE and BAPS at $K = 2$. In this two-dimensional plot, the S mainland populations, as well as the SE ones and the Island populations (Minorca, Majorca and the Canary Islands) were well differentiated along the axis 2, which accumulated 26.2% of variance.

The NJ tree separated all SE mainland populations, Minorca and the Majorcan population of BONA from the remaining populations, in a highly supported group (78% bootstrap support (BS) value; Fig. 6A). A similar NJ tree was retrieved when the admixed individuals indicated by STRUCTURE were excluded (Fig. 6B). The remaining sub-divisions found in the NJ trees correspond mainly to the populations sampled although BS values were always very low, with or without admixed individuals (< 43%, Figs. 6A and 6B).

## Genetic differentiation and isolation

Overall, genetic differentiation was significantly high (AMOVA FST = 0.748, $P < 0.00001$). The analysis performed over the 19 populations sampled indicated that
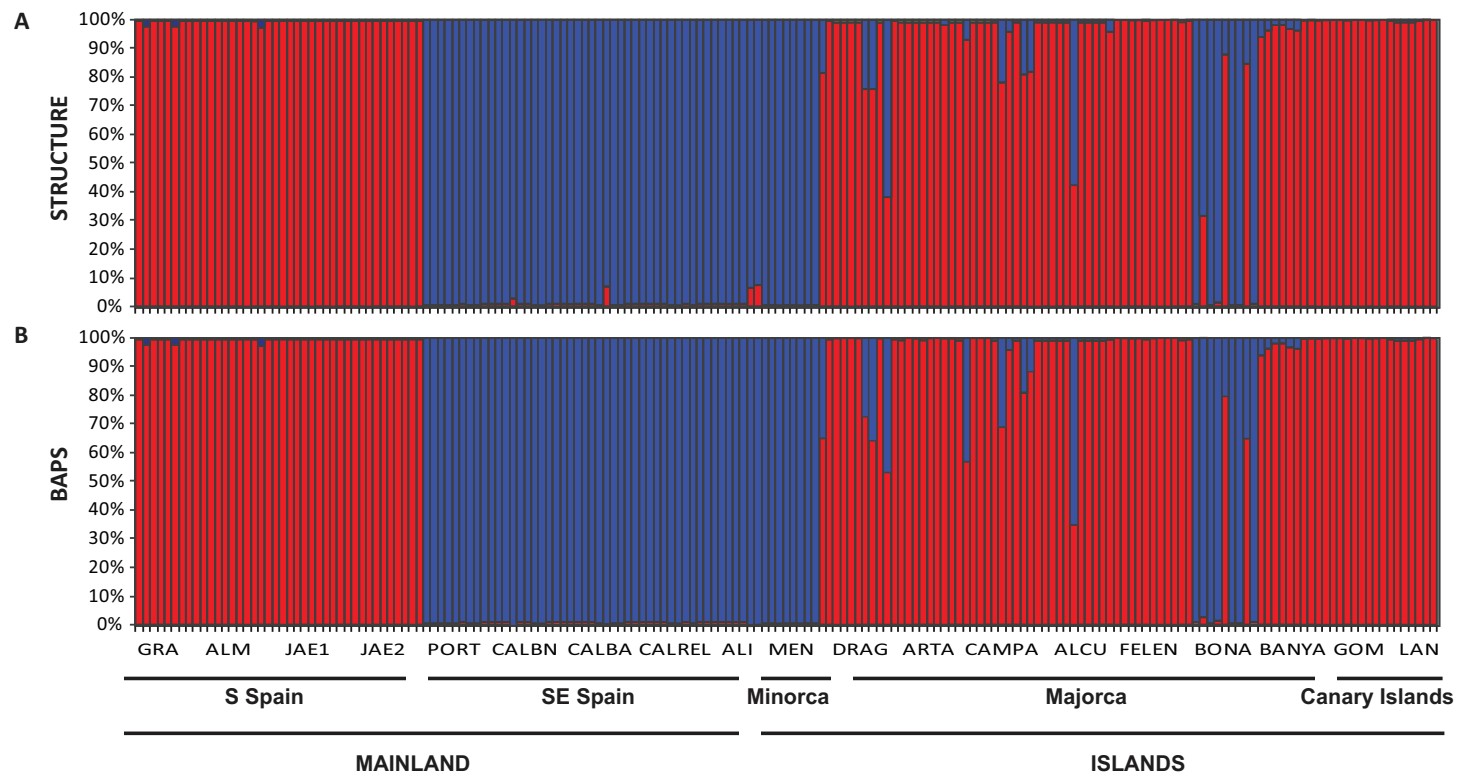

**Figure 4 Population structure of *Brachypodium stacei*.** Population structure of 181 individuals of *Brachypodium stacei* based on 10 nSSRs and using the best assignment result ($K = 2$) retrieved by STRUCTURE (A) and by BAPS (B) with $K$ from 1 to 10 (replicated 10×) under an admixture model. Each individual is represented by a thin horizontal line divided into $K$ colored segments that represent the individual's estimated membership fractions in $K$ clusters. The different geographic areas are labelled below the graph. Abbreviations of populations follow those indicated in Table 1.

75 and 25% of the genetic variation was attributed to variation among and within populations, respectively ($P < 0.00001$; Table 4). When analyzing the two genetic groups retrieved by STRUCTURE, an independent AMOVA attributed 24, 54 and 22% of the total variation to variation among groups, among populations within groups and within populations (Table 4). Fixation indices of this analysis were $F_{ST} = 0.779$, $F_{SC} = 0.710$ and $F_{CT} = 0.240$. To further investigate genetic differentiation between mainland and island populations, an independent AMOVA also attributed the highest percentage of variation among populations within groups (68% of the total variance; $F_{ST} = 0.758$, $F_{SC} = 0.737$ and $F_{CT} = 0.077$; Table 4). However, genetic variation was equally partitioned among groups, among populations within groups and within groups when analyzing only island populations ($F_{ST} = 0.672$, $F_{SC} = 0.516$ and $F_{CT} = 0.322$), and predominant among groups and among populations within groups when analyzing only mainland populations ($F_{ST} = 0.884$, $F_{SC} = 0.783$ and $F_{CT} = 0.464$; Table 4).

The Mantel's test did not detect any significant correlation between the genetic distance $[F_{ST}/(1 - F_{ST})]$ and the geographical distance of the populations studied here ($r^2 = 0.83$, $y = 1.755x - 0.223$, $P = 0.085$).

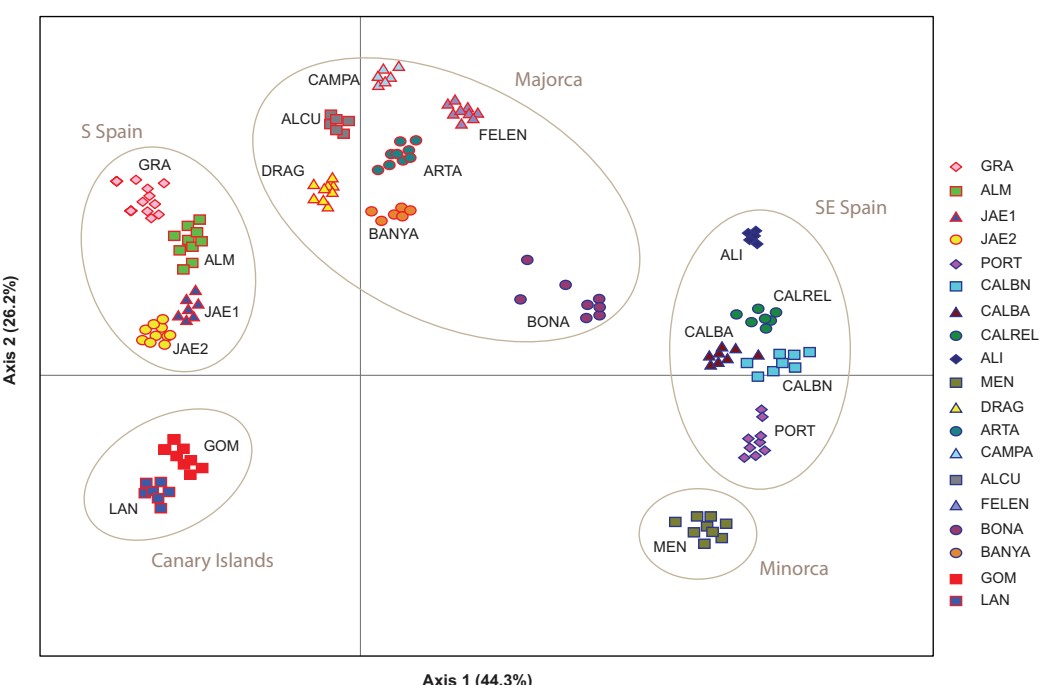

**Figure 5 Genetic relationships among *Brachypodium stacei* populations based on Nei's genetic distance.** Principal Coordinate analysis (PCoA) samples using the scored nSSRs markers. Percentage of explained variance of each axis is given in parentheses. Population symbols are shown in the chart.

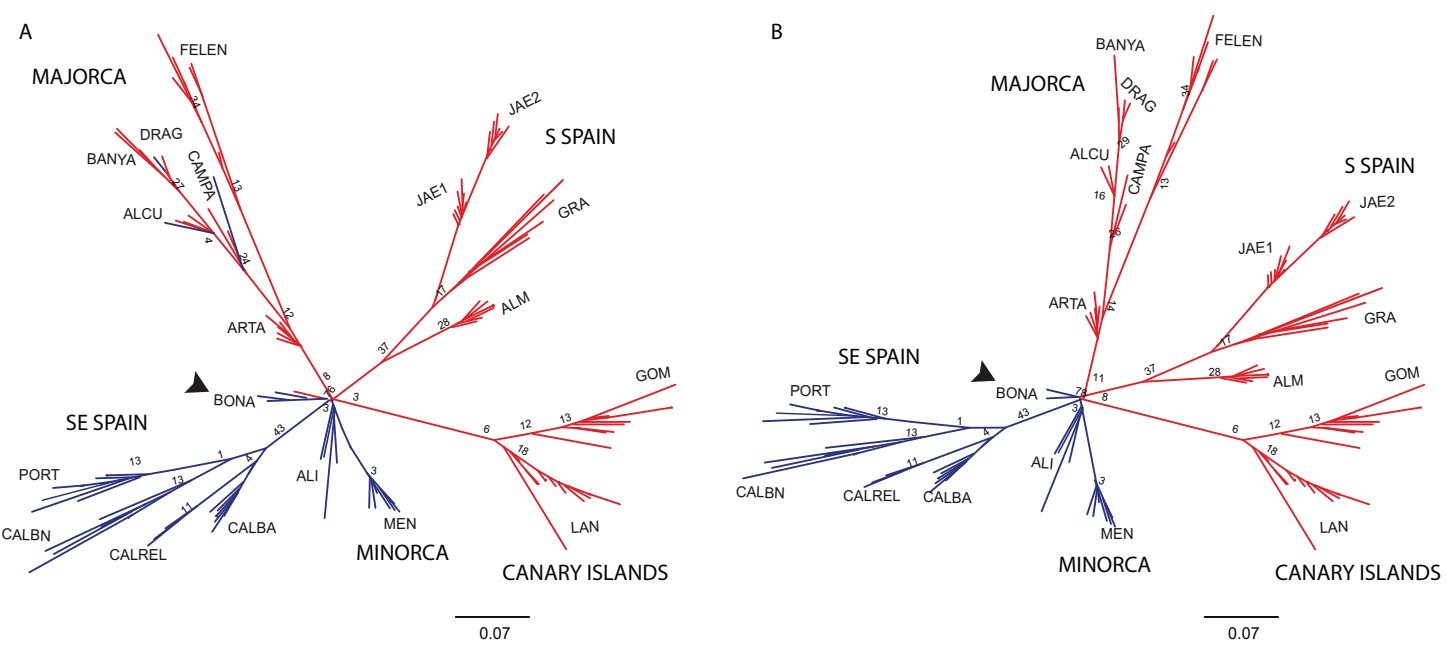

**Figure 6 Unrooted neighbor-joining trees of *Brachypodium stacei* populations based on Nei's genetic distance.** Unrooted neighbor-joining tree showing relationships among the individuals collected in 19 populations. Numbers associated with branches indicate bootstrap values based on 1,000 replications. Colours followed the ones depicted in Fig. 4 for *K* = 2. Population codes are indicated in Table 1. (A) Genetic relationships among all individuals of *B. stacei*. (B) Genetic relationships without the individuals of *B. stacei* showing admixture in STRUCTURE. Note that the Majorcan population of BONA (arrow) is grouped with SE mainland populations in both NJ trees.

**Table 4 Analysis of molecular variance (AMOVA) for 19 populations of *Brachypodium stacei*.**

|  | Source of variance | d.f. | Variance components | % Variance |
|---|---|---|---|---|
| All populations | Among populations | 18 | 1.011 | 74.88 |
|  | Within populations | 343 | 0.339 | 25.12 |
| Between genetic groups defined by STRUCTURE and BAPS (*K* = 2) | Among groups | 1 | 0.370 | 24.05 |
|  | Among populations within groups | 17 | 0.831 | 53.94 |
|  | Within populations | 343 | 0.339 | 22.01 |
| Mainland vs. islands | Among groups | 1 | 0.109 | 7.77 |
|  | Among populations within groups | 17 | 0.954 | 68.04 |
|  | Within populations | 343 | 0.339 | 24.19 |
| Within mainland populations (S Spain vs. SE Spain) | Among groups | 1 | 0.831 | 46.41 |
|  | Among populations within groups | 7 | 0.752 | 41.99 |
|  | Within populations | 161 | 0.207 | 11.60 |
| Within island populations (Balearic islands vs. Canary islands) | Among groups | 2 | 0.448 | 32.26 |
|  | Among populations within groups | 7 | 0.486 | 34.97 |
|  | Within populations | 182 | 0.455 | 32.77 |

## DISCUSSION

### Incidence of a highly selfing mating system

From our genetic study, selfing rates of *B. stacei* were estimated as 79% across all populations, even reaching values as high as 95% in some populations (Table 1) although CI values were fairly wide. Even if null alleles have not been detected, the high variation occurring between loci for many of the genetic parameters estimated (Table 3) might influence the reported genetic values. Nonetheless, there was clearly a predominance of homozygous individuals, suggesting that *B. stacei* is primarily a selfing species like its close congener *B. distachyon* (*Draper et al., 2001*; *Gordon et al., 2014*). Although the respective ancestors of these two annual species likely split 16.2 (*B. stacei*) and 10.2 (*B. distachyon*) Mya ago (cf. *Catalán et al., 2016b*), species divergence was not followed by changes in the mating system of *B. distachyon* and *B. stacei*. This analogous mating system is consistent with similarities in floral morphology and floral structure in the two species since they both bear relatively small (mean 7–9 mm) cleistogamous or cleistogamous-type florets having minute (< 0.8 mm) non-exerted anthers (*Catalán et al., 2016a*). Pollination in *B. distachyon* usually occurs in closed flowers leading to extremely high levels of homozygosity (*Vogel et al., 2009*), such as the ones reported here for *B. stacei*. Even more recently diverged species, such as the perennial *B. sylvaticum*, display a predominantly selfing system, although the levels of heterozygosity suggest that this species outcross more often than *B. distachyon* and *B. stacei* (*Steinwand et al., 2013*).

In nature, selfing is thought to be favored due to its inherent transmission advantage, as well as assuring reproduction when pollinators or available mates are scarce (*Marques, Draper & Iriondo, 2014*) and it is expected to evolve whenever these advantages outweigh the costs of inbreeding depression (*Charlesworth & Willis, 2009*).

But contrary to these short-term benefits, selfing might also reduce effective recombination rate leading to frequent genetic bottlenecks (*Goldberg et al., 2010*). Recombination is generally thought to be advantageous because it breaks down associations between alleles (linkage disequilibrium), which might lead to the fixation of deleterious mutations (*Charlesworth & Charlesworth, 2000*). As a result, it has long been argued that the evolutionary potential of highly selfing species is quite limited as a result of reduced genetic diversity and recombination rates (*Lynch, Conery & Burger, 1995*). However, many important crops such as wheat, barley, beans, and tomatoes, are predominantly self-fertilizing species despite the possibility of linkage drag (*Morrell et al., 2005*). Likewise, linkage disequilibrium is absent in *B. stacei* despite being a highly selfing species. There are several possible explanations. The first is that the relatively low levels of linkage disequilibrium results from a recent transition from a strict outcrossing ancestral mating system to a predominantly selfing one, so that the recombination events would still be present (*Lin, Brown & Clegg, 2001*). However, recent phylogenetic studies indicated that *B. stacei* is the earliest extant diverging lineage within the genus and that other early splits also resulted in selfing species (e.g., *B. mexicanum*; *Catalán et al., 2016b*). The second possible explanation is that in a temporal time scale of more than 38 Mya to the common ancestor of the *Brachypodium* stem node (e.g., MRCA of the Brachypodieae/core pooids split; *Catalán et al., 2016b*), even a very low number of outcrossing events might be enough to promote a certain level of recombination. Although plant species might usually mate through selfing, few are strictly selfing (*Igic & Busch, 2013*), creating opportunities for recombination that helps to break down associations between alleles. Large population sizes, which are not uncommon in *B. stacei*, might also reduce linkage disequilibrium. For instance, the near worldwide-distributed *Arabidopsis thaliana* is predominantly a selfing annual species but exhibits a rapid decay in linkage drag in several populations (*Nordborg et al., 2005*).

## Origin of island populations

Many plant phylogeographic studies have concluded that genetic diversity erodes across colonization steps, but islands usually exhibit high frequencies of endemism in comparison with large continental areas as a consequence of isolation and habitat diversity (*Kim et al., 1996*; *Sanmartín, van der Mark & Ronquist, 2008*; *Vitales et al., 2014*; see review in *Caujapé-Castells (2011)*). In our study, the genetic structure retrieved by the Bayesian analyses of STRUCTURE or BAPS, or by the results retrieved from the PCoA and the NJ tree suggests a scenario of colonization from the mainland Iberian Peninsula into the islands. Individuals collected in Minorca clustered with SE mainland Iberian populations, whereas individuals from the Canary Islands and most of the Majorcan populations were clustered with S mainland populations (with the exception of BONA which is more related to the SE mainland populations). The large number of alleles (89%) shared between the individuals collected in the Canary Islands and the ones collected in S Spain could support the hypothesis of colonization from the mainland Iberian Peninsula. A recent colonization scenario from a mainland

Iberian source fits well the plausible origin of the oceanic Canary island populations, which show low levels of genetic diversity and multilocus genetic profiles that are a subset of those found in S Spain (Table 1; Fig. 2). Additionally, Canarian populations of *B. stacei* have shown to be phenotypically close to S Spain populations (D. López-Alvarez, P. Catalán, 2016, unpublished data). However, islands could have also been colonized by North African coastal populations of *B. stacei* since ecological niche models predict the existence of conditions suitable for the existence of this species in that area (*López-Alvarez et al., 2015*).

Single vs. multiple colonization scenarios from mainland Iberian sources have been proposed to explain the origins of the Macaronesian plant populations (cf. *Díaz-Pérez et al., 2012*); however, most of them gave rise to new species (*Kim et al., 1996*; *Francisco-Ortega, Jansen & Santos-Guerra, 1996*; *Francisco-Ortega et al., 1997*; *Caujapé-Castells, 2011*). Even if *B. stacei* grows preferentially in relatively stable shady coastal and lowland places along its distribution area (*Catalán et al., 2016a*), seeds of this annual species could be also occasionally dispersed through long distances, as inferred from genetic studies (*López-Alvarez et al., 2012*). The fact that all the studied individuals of the Canarian GOM and LAN populations are morphologically similar to those of the remaining Mediterranean populations (*Catalán et al., 2016a*) indicates that they belong to the same species, suggesting that the introduction of the plant in the Canary isles was probably a very recent one.

Contrastingly, the Balearic populations of *B. stacei* show similar, or even higher genetic diversity values in the case of the Majorcan populations (e.g., BONA, Majorca; Table 1), than the mainland Iberian populations. This scenario could be explained by old colonization events from the mainland followed by insular isolation, which might have favored the appearance and accumulation of new allelic variants and genotypes along time (Fig. 2). Also, admixture after multiple colonization's could have contributed to this scenario, which has been reported to have occurred in other postglacial recolonizations in Europe (e.g., *Lexer et al., 2010*; *Krojerová-Prokešová, Barančeková & Koubek, 2015*). The palaeogeographic configuration of the continental Balearic Islands could have facilitated the migration of coastal SE Spain and S Spain *B. stacei* populations into Minorca and Majorca, and the repeated colonization (and admixture) of the later island from multiple continental sources (Fig. 4). The southern Iberian region together with its eastern Iberian range, the Balearic isles and Provence formed a continuous geological region that split into several microplates during the Oligocene (*Cohen, 1980*). In the late Oligocene (30–28 Ma) the Balearic microplate separated from the eastern proto-Iberian peninsula (*Cohen, 1980*; *Rosenbaum, Lister & Duboz, 2002*) but during the Messinian drought and salinity crisis of the Mediterranean in the late Miocene (c. 6–5 Ma), the Balearic islands formed a single land mass (*Gautier et al., 1994*) and several land bridges re-established the connection with the eastern Iberian Peninsula (*Lalueza-Fox et al., 2005*). Even after the opening of the Gibraltar strait and the refilling of the Mediterranean basin (c. 5 Ma), several land bridges were again created during Middle-Upper Pleistocene that connected the Balearic Gymnesian isles between themselves and between those

islands and the mainland eastern Iberian Peninsula (c. 0.40 Ma; *Cuerda, 1975*), favoring the colonization of the islands from mainland plant populations stocks (*Garnatje et al., 2013*).

## Is there a role for ecogeographical isolation in *Brachypodium*?

High values of genetic differentiation and a signature of strong genetic structure were found in *B. stacei*. Though genetic differentiation values obtained for other selfing but more outcrossing species of *Brachypodium* are relatively high (e.g., *B. sylvaticum*: $F_{ST}$ 0.480 ± 0.28 in native Eurasian populations, and 0.446 ± 0.26 in invasive western North American populations; *Rosenthal, Ramakrishnan & Cruzan, 2008*), the high values of genetic differentiation found in *B. stacei* are puzzling. Lower genetic differentiation values than the ones found here usually correspond to different grass species (e.g., *Festuca*, *Díaz-Pérez et al., 2008*). However, all *B. stacei* individuals examined in this study are morphologically similar and correspond to what is considered to be the same species (*Catalán et al., 2016a*). Population turnover is expected to increase genetic differentiation among populations if colonizers are dispersed from different sources (*Pannell & Charlesworth, 2000*). That might only be true for *B. stacei* if wind or other vectors are dispersing seeds across populations, as hypothesized for the also annual and autogamous *B. distachyon* (*Vogel et al., 2009*; *Mur et al., 2011*). That would probably erase the patterns of genetic structure that we have found in STRUCTURE, BAPS, the NJ tree and the PCoA analyses (Figs. 3–5), though the high rates of selfing observed could explain the high levels of genetic differentiation and strong population structure of *B. stacei*, like reported in other primarily selfing plants (*Nybom, 2004*).

Because plants are sessile they experience generations of selection that result in adaptive genetic differentiation to local environmental conditions if there is a strong pressure (*Kremer, Potts & Delzon, 2014*). Although we have no empirical information for *B. stacei*, the distribution of the close relatives, *B. distachyon* and its allopolyploid derivative *B. hybridum*, indicates that they are geographically structured in mesic to arid environments, with *B. distachyon* occurring predominantly in more mesic sites and *B. hybridum* in more aridic sites (*Manzaneda et al., 2012*). *Brachypodium hybridum* is also more efficient in its water usage being significantly more tolerant to drought than *B. distachyon* and behaving as a drought-escapist (*Manzaneda et al., 2015*). Also, environmental niche model analyses indicate a preference of *B. stacei* for warm and arid Mediterranean places (*López-Alvarez et al., 2015*), though its habitat preferences are for shady places, probably as a protection from direct insulation in the aridic environment (*Catalán et al., 2016a*). Therefore, all together, results suggest an important role for ecogeographical differentiation in these lineages of *Brachypodium* (*Manzaneda et al., 2012*; *Manzaneda et al., 2015*; *López-Alvarez et al., 2015*; *Catalán et al., 2016a*; *Catalán et al., 2016b*). More detailed ecological studies are necessary to understand the potential ecological tolerance of *B. stacei* to the arid conditions.

## Perspectives towards new genomics initiatives in *B. stacei*

The ongoing de-novo genome sequencing of *B. stacei* led by the Joint Genome Institute and the International *Brachypodium* Consortium (http://jgi.doe.gov/our-science/science-programs/plant-genomics/brachypodium/) will provide significant insights into the mechanisms of polyploid hybrid speciation within the complex *B. distachyon–B. stacei–B. hybridum*, also allowing comparative studies of genomics and development of functional traits in other crop plants. Biological features, such as a selfing system, a diploid genome and having amenable growing conditions are all advantages for the development of a model system and for genomic resources. All seem to be present in *B. stacei*. Previous studies showed that the species is diploid (*Catalán et al., 2012*) and can easily grow even in laboratory conditions, germinating in less than one week like we have seen in our own laboratory (P. Catalán, 2016, unpublished data).

The results from the present study support the existence of a highly selfing system, which from a practical perspective is an advantage in a model species since it simplifies the process of obtaining pure lines under laboratory conditions (*Gordon et al., 2014*). But plant breeding requires the presence of genetic variability in order to increase the frequencies of favorable alleles and genetic combinations. Populations from SE Spain are genetically different from the ones in S Spain and further differentiation might occur in the islands especially in Majorca and Minorca, where several unique alleles were found. Future studies need to test if population differentiation reflects local adaptation to different environments. Nonetheless, researchers of GWA studies need to be careful to avoid reporting false positive signals (i.e., identifying loci that are not responsible for the variation in the trait), which can be caused by population structure (*Platt, Vilhjálmsson & Nordborg, 2010*; *Brachi, Morris & Borevitz, 2011*). In this sense, several efforts have been raised to address this problem statistically (*Pritchard et al., 2000*; *Price et al., 2006*; *Yu et al., 2006*) and recent GWAS can detect loci that are involved in the natural variation of traits even in highly structure plants like *Arabidopsis* (*Nordborg et al., 2005*).

Thus, to help future genomic initiatives involving *B. stacei* we recommend the following guidelines: (1) a collection of different accessions reflecting different ecological pressures should be generated in order to recover the full genomic diversity of *B. stacei*; (2) the creation of a gene bank collection of these materials constitutes a practical and useful reservoir of genetic variation to avoid uniform cultivars and genetic erosion; (3) collections should be accessible to facilitate the interchange of material useful for breeding and other studies. Finally, there is a lack of information for other areas of the Mediterranean, especially the Eastern Mediterranean–SW Asian area, where *B. stacei* has been also found (*López-Alvarez et al., 2012*; *López-Alvarez et al., 2015*; *Catalán et al., 2016a*). A comprehensive study including populations from other Mediterranean areas is compulsory to fully discover the phylogeographic patterns and genetic diversity of *B. stacei*.

## ACKNOWLEDGEMENTS

We thank the Spanish Centro de Recursos Fitogenéticos (CRF-INIA), Consuelo Soler and Antonio Manzaneda for providing us some *B. stacei* seeds, Maria Luisa López-Herranz and Diana López-Alvarez for laboratory and greenhouse assistance, and William Scott for linguistic assistance.

### Funding

The study has been funded by a Spanish Ministry of Science grant project (CGL2012-39953-C02-01). IM received funding from the People Programme (Marie Curie Actions) of the European Union's Seventh Framework Programme (FP7/2007–2013) under REA grant agreement PIOF-GA-2011-301257. VS was funded by a Tomsk State University (TSU, Russia) PhD fellowship. PC and IM were partially funded by a Bioflora grant cofunded by the Spanish Aragon Government and the European Social Fund. The funders had no role in study design, data collection and analysis, decision to publish, or preparation of the manuscript.

### Grant Disclosures

The following grant information was disclosed by the authors:
Spanish Ministry of Science grant project: CGL2012-39953-C02-01.
European Union's Seventh Framework Programme: FP7/2007–2013.
REA grant agreement: PIOF-GA-2011-301257.

### Competing Interests

The authors declare that they have no competing interests.

### Author Contributions

- Valeriia Shiposha performed the experiments, analyzed the data, wrote the paper, prepared figures and/or tables.
- Pilar Catalán conceived and designed the experiments, analyzed the data, contributed reagents/materials/analysis tools, wrote the paper, reviewed drafts of the paper.
- Marina Olonova wrote the paper.
- Isabel Marques performed the experiments, analyzed the data, wrote the paper, prepared figures and/or tables, reviewed drafts of the paper.

### Data Deposition

The raw data has been supplied as Supplemental Dataset Files.

### Supplemental Information

Supplemental information for this article can be found online at http://dx.doi.org/10.7717/peerj.2407#supplemental-information.

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
