# Peer review of "Genetic structure and diversity of the selfing model grass Brachypodium stacei (Poaceae) in Western Mediterranean: out of the Iberian Peninsula and into the islands"

_PeerJ, doi:10.7717/peerj.2407_

## Round 0.1 · original submission · Minor Revisions

This manuscript is a nice study that will be of interest to the Brachy community as well as being an interesting biogeographic work in its own right. The reviewers have returned only a few major (and several minor) points, most of which the authors should be able to address. It will be particularly helpful to clarify the three hypotheses (ln 294-307) discussed and especially the third, which the authors favour; the rationale behind this was not clear to me as well as to one reviewer.

Reconsideration, rephrasing, or deletion of that section would be helpful. Particularly how might it be that 'that deleterious alleles are rapidly erased in selfing plants'? I took a look at the Shimizu et al review and it's not clear to me exactly what it is the authors are referring to. I may have missed an important point, but the statement seems highly improbable, so if the authors can point to clearer evidence, it would be helpful. Other than this, the manuscript was very clear and well expressed. Minor revisions are recommended, either by providing additional details/analyses and/or explaining why these are not feasible/necessary.

Reviewer 1 ·

Basic reporting

The paper is well-written with appropriate intro and background, and conforms to the PeerJ standard. A few of the figures could be improved for clarity (see comments below).

Experimental design

The experiment seems well-motivated and conducted and the methods are well described.

Validity of the findings

The findings seem valid, however the authors need to report and discuss any uncertainty around their estimates of heterozygosity, Fis, and selfing rate (see comments below). In addition, I the raw genetic data is not included in the paper and it does not seem to have been submitted to a repository.

Additional comments

1) The authors should provide confidence intervals around their estimates of heterozygosity, Fis, and selfing rate, especially because there is high variation between loci for many of the parameters estimated (see Table 3).

2) The explanations given for why there is low LD in B. stacei despite a high selfing rate are not very convincing. Hypothesis 3 seems to consist of two hypotheses. The first one of these, that selfers may have higher recombination rates, does not seem applicable here: selfers have high LD because high homozygosity prevents recombination from breaking up LD, and this process won’t be mitigated by higher recombination rates. The second piece of hypothesis 3 seems vague: it’s unclear to me why purging of deleterious alleles in selfers has anything to do with LD. Ultimately, it does not seem that surprising that there is low LD in this system since there is still a fair amount of outcrossing and the loci investigated are likely far apart and/or on different chromosomes.

3) Figure 5 is difficult to link with other data -- it would be helpful to label which populations belong to island/mainland and S/SE, etc.

4) It’s unclear how individuals with mixed blue/red ancestry are displayed on Figure 6.

5) The authors write that the observed proportion of samples with unique multi-locus genotypes is lower than expected (line 210), but it’s unclear where this expectation comes from.

6) As an aside, observations of high population structure across the range of B. stacei do not seem encouraging for future GWAS in this system since structure can cause false-positives in GWAS when not controlled for.


Typos:
Line 269: “homizygous” -> “homozygous”
Line 272: “Ma ago” -> “Mya ago”

·

Basic reporting

The study brings novel and interesting results on genetic structure of potentially important plant species and it is generally well suitable for publication in PeerJ.

Experimental design

The study meets the requirements of PeerJ

Validity of the findings

The study meets the requirements of PeerJ

Additional comments

I have one major methodological and one interpretational concerns, in addition to few minor points.

1) The authors used cross-amplified SSR loci from a relatively divergent species, thus rising question of presence of null alleles. This is particularly relevant in cases of inference of selfing rates. In order to improve relevance of the interesting results, I thus recommend performing an additional, complementary, analysis allowing simultaneous inference of inbreeding coefficient and null alleles in a Bayesian framework, e.g., impemented in INEST (Chybicki IJ, Burczyk J (2009) Simultaneous estimation of null alleles and inbreeding coefficients. Journal of Heredity 100: 106-113).

2) lines 324-329: I consider the discussion of Iberian origin of the Canary populations very speculative, as the samples from N Africa (a frequent source of Macaronesian diversity) are lacking. The proportion of shared alleles with Iberian Pen. is not convincing evidence, as the same may be true for the (missing) N African samples. On the other hand, (not obligatory!) the Discussion might be enriched by referring to the habitats occupied by B. stacei in the Canaries (strict preference of anthropogeneous sites would, e.g., support its recent introduction by humans).

Minors:
* * *
I wonder why the authors do not consider admixture as a potential (non-exclusive) explanation of the high diversity of the Majorcan populations. In particular for pop. BONA, it is indicated by its interemdiate position in the PCA and admixture in Structure). Admixture after multiple colonization is a well-known phenomenon in European phylogeographies, particularly in relation with postglacial re-colonization, but might be interesting novel finding in Mediterranean islands.

Fig. 6 (NJ tree): Tree is not a suitable visualisation method for reticulate patterns such as under potential admixture (see above). I suggest presenting an additional NJ tree where the potentially admixed indivs. (as indicated by Structure) were excluded and/or displaying the distances of the whole dataset using network (e.g., in SplitsTree).

Fig. 1 (map) it would be useful to plot/border the two major Structure groups directly into this map, to better visually link the genetic grouping with the geographic distributions (e.g using pie charts).

l. 269 homozYgous

---

## Round 0.2 · Minor Revisions

Thank you for the much improved manuscript. If the authors can meet the comments of the reviewer I foresee no reason this should not be published.

A primary concern remaining is point 1 by the reviewer ("The claim that this study shows local adaptation (line 429) is not at all supported by the data. While there is genetic variation at microsatellites, microsatellites are generally assumed to be neutral and there is no indication that they are not neutral here. It would be fairer to state that there is potential for local adaptation in B. stacei. (Apologies for missing this in the first review)"). This issue should be met easily with editing.

Reviewer 1 ·

Basic reporting

The study meets the requirements of PeerJ

Experimental design

The study meets the requirements of PeerJ

Validity of the findings

The study meets the requirements of PeerJ

Additional comments

In general, the authors have satisfactorily responded to my comments and made appropriate changes.

I have a few minor comments below,

1. The claim that this study shows local adaptation (line 429) is not at all supported by the data. While there is genetic variation at microsatellites, microsatellites are generally assumed to be neutral and there is no indication that they are not neutral here. It would be fairer to state that there is potential for local adaptation in B. stacei. (Apologies for missing this in the first review).

2. You could cite Goldberg et al. 2010 Science for more evidence that selfing can be deleterious (line 308)

3. There are some typos in the manuscript -- I’ve listed the ones I found below but it could use a thorough proofreading.

Line 46: Besides its -> Besides their
Line 117: manufactures’ -> manufacturer’s
Figure 2 X axis label: Alelles -> Alleles
Legend for Figure 6: braches -> branches

·

Basic reporting

I already reviewed a previous version of this ms. I found this version much improved and the concerns brought forward by the reviewers were adequately addressed.

Experimental design

OK

Validity of the findings

OK

Additional comments

OK

---

## Round 0.3 · accepted · Accept

Thank you for choosing to publish in PeerJ, I am happy to see you did. I'm confident that your manuscript will be of broad interest to the Brachy community, but also to anyone studying the evolution and ecology of wild plant populations.

Reviewer 1 ·

Basic reporting

The study meets the requirements of PeerJ

Experimental design

The study meets the requirements of PeerJ

Validity of the findings

The study meets the requirements of PeerJ

Additional comments

The authors have responded satisfactorily to my suggestions.